circMSH3 is a potential biomarker for the diagnosis of colorectal cancer and affects the distant metastasis of colorectal cancer

Shen Jian 1
Min Yu 2 3
Luo Jingen 4
Tang Xingkui 4
Han Zeping 1
Luo Wenfeng 1
Xie Fangmei 1
Cao Mingrong 5
Zhou Taicheng zhoutaicheng@126.com 6
He Jinhua 332518579@qq.com 1
1 Department of Central Laboratory, Guangzhou Panyu Central Hospital , Guangzhou , Guangdong , China
2 Guangzhou University of Chinese Medicine , Guangzhou , Guangdong , China
3 Department of Rehabilitation Medicine, Guangzhou Panyu Central Hospital , Guangzhou , Guangdong , China
4 Department of General Surgery, Guangzhou Panyu Central Hospital , Guangzhou , Guangdong , China
5 Department of Hepatobiliary Surgery, The First Affiliated Hospital of Jinan University , Guangzhou , Guangdong , China
6 Department of Gastroenterological Surgery and Hernia Center & Guangdong Provincial Key Laboratory of Colorectal and Pelvic Floor Diseases, The Sixth Affiliated Hospital, Sun Yat-sen University , Guangzhou , Guangdong , China
Amdare Nitin
Electronic publication date: 2023 Nov 7
Publication date: 2023
Volume: 11
Electronic Location ID: e16297
Received 2023 Apr 19; Accepted 2023 Sep 23
Copyright: ©2023 Shen et al.
Copyright year: 2023
Copyright holder: Shen et al.
License: This is an open access article distributed under the terms of the Creative Commons Attribution License, which permits unrestricted use, distribution, reproduction and adaptation in any medium and for any purpose provided that it is properly attributed. For attribution, the original author(s), title, publication source (PeerJ) and either DOI or URL of the article must be cited.
License URL: https://creativecommons.org/licenses/by/4.0/

Keywords: Colorectal cancer, Hsa_circ_0003761, CircMSH3, RNA-protein pull-down, MiR-1276, MiR-942-5p, MiR-409-3p

Funding: The Basic and Applied Basic Research Foundation of Guangdong Province 2022A1515220217 The Medical Scientific Research Foundation of Guangdong Province A2023216 A2022524 The Science and Technology Program of Guangzhou 202201010840 202201010810 202102080532 202002030032 202002020023 The Health Commission Program of Guangzhou 20212A010025 20201A010085 The Science and Technology Project of Panyu, Guangzhou 2022-Z04-009 2022-Z04-090 2022-Z04-072 2021-Z04-053 2020-Z04-026 2019-Z04-02 The Scientific Research project of Guangzhou Panyu Central Hospital 2022Y002 2021Y004 2021Y002 The Guangdong Basic and Applied Basic Research Fund Enterprise Joint Fund 2022A1515220217 The Guangdong Provincial Medical Research Fund A2023216 This work was supported by the Basic and Applied Basic Research Foundation of Guangdong Province (No. 2022A1515220217), the Medical Scientific Research Foundation of Guangdong Province (No. A2023216; A2022524), the Science and Technology Program of Guangzhou (No. 202201010840; 202201010810; 202102080532; 202002030032; 202002020023), the Health Commission Program of Guangzhou (20212A010025; 20201A010085), the Science and Technology Project of Panyu, Guangzhou (2022-Z04-009; 2022-Z04-090; 2022-Z04-072; 2021-Z04-053; 2020-Z04-026; 2019-Z04-02), the Scientific Research project of Guangzhou Panyu Central Hospital (No. 2022Y002; 2021Y004; 2021Y002) the Guangdong Basic and Applied Basic Research Fund Enterprise Joint Fund (2022A1515220217) and the Guangdong Provincial Medical Research Fund (A2023216). The funders had no role in study design, data collection and analysis, decision to publish, or preparation of the manuscript.

==============================
Objectives

To identify the most significantly differentially expressed circular RNAs (circRNAs) in colorectal cancer (CRC) tissues in terms of their expression levels and circularity, and to analyze the relationship between their expression levels and the clinical characteristics of patients.

Methods

circRNA RNA-seq technology was used to screen differentially expressed circRNAs in CRC. Sanger sequencing was used to identify circRNA back-splice junction sites. The relative expression levels of hsa_circ_0003761 (circMSH3) in CRC tissues and cell lines were detected by quantitative real-time fluorescence PCR technology. An RNA-protein pull-down assay was used to detect protein binding to circRNAs. Dual-luciferase reporter gene vectors were constructed to verify that circRNAs bind to microRNAs.

Results

Four hundred twenty circRNAs were found to be upregulated, and 616 circRNAs were downregulated. circMSH3 was derived from the MutS homolog 3 (MSH3) gene and was formed by a loop of exons 9, 10, 11, and 12. In 110 pairs of CRC and adjacent tissues, circMSH3 expression was 4.487-fold higher in CRC tissues. circMSH3 was also highly expressed in the HT-29 and LOVO CRC cell lines. The expression level of circMSH3 was associated with distant metastasis in CRC patients (P = 0.043); the area under the curve (AUC) of circMSH3 for CRC diagnosis was 0.75, with a sensitivity and specificity of 70.9% and 66.4%, respectively. circMSH3 could bind to a variety of proteins, mainly those involved in RNA transcription, splicing, cell cycle, and cell junctions. Furthermore, circMSH3 could bind to miR-1276, miR-942-5p, and miR-409-3p.

Conclusion

circMSH3 is a potential biomarker for the diagnosis of CRC and affects the distant metastasis of CRC. Multiple RNA-binding protein binds to circMSH3, and circMSH3 binds to miR-1276, miR-942-5p, and miR-409-3p, thereby affecting the expression of circMSH3.

Introduction

Circular RNAs (circRNAs) are a newly discovered class of endogenous non-coding RNAs that form special covalently closed circular structures without 5′–3′ polarity and a polyadenylation tail. circRNAs are involved in almost all cellular physiological processes, and the abnormal expression of circRNAs is closely related to tumor invasion and metastasis. circRNAs are widely expressed in tissues and blood, are highly stable, and have high specificity (Chen & Shan, 2021; Zhang et al., 2021; Liu & Chen, 2022). Therefore, circRNAs can be used as potential biomarkers and molecular targets for the diagnosis and treatment of malignant tumors.

Colorectal cancer (CRC) is one of the most common malignant tumors of the digestive tract worldwide and is a serious threat to human life and health (Enblad, Graf & Birgisson, 2018; Sung et al., 2021). The occurrence and development of CRC is a multi-step, multi-stage, and multi-gene process, in which abnormal gene expression plays an important role (Segelman et al., 2012; Vuik et al., 2019). Although considerable research has been performed on CRC, its pathogenic mechanism has not been elucidated (Koppe et al., 2006; Lemmens et al., 2011). Therefore, further in-depth research on the pathogenesis of CRC is needed to provide a basis for its early diagnosis, prognosis, and efficacious treatment. At present, studies have shown that circRNAs play an important role in tumorigenesis and development (Fang et al., 2019; Arnaiz et al., 2019; Kristensen et al., 2022), but the key circRNAs that regulate CRC progression and their molecular regulatory mechanisms remain unclear. Therefore, identifying the key circRNAs in CRC and elucidating their functions and molecular mechanisms of action will provide new molecular markers for the clinical diagnosis and treatment of CRC as well as provide a scientific basis for the design of therapeutic strategies to inhibit metastasis.

Material and Methods

Specimen source

Tumor and paracancerous tissues were collected from 110 patients with colon cancer who underwent surgical resection in the Central Hospital of Panyu District, Guangzhou. Prior to this, none of the patients had received neoadjuvant chemotherapy, radiotherapy, or other antitumor therapy, and all tissue samples were pathologically confirmed. All patients or their families signed an informed consent form to participate. The study was approved by the Ethics Committee of the Guangzhou Panyu District Central Hospital, approval number PYRC-2023-017.

Cell culture

CRC cell lines (HCoEpiC,HCT15,SW116,RKO,Colo205,HCT116,Caco2,Lovo, HT-29,SW480,SW620 and DLD-1) were purchased from the Shanghai Institute of Cell Biology (Shanghai, China). The cell lines were grown in Dulbecco’s modified Eagle’s medium/F12 medium containing 10% fetal bovine serum and maintained at 37 °C in a humidified incubator with 5% CO2. The medium was changed every other day. The cells were digested with 0.25% trypsin and subcultured, and cells with logarithmic growth and 95% viability were selected for further experiments (Itoh et al., 2002; Bachmayr-Heyda et al., 2015; Lu et al., 2017; Xu et al., 2020; Li & Li, 2021).

circRNA sequencing (circRNA-seq)

Selected 10 pairs of colorectal cancer patients’ cancer tissue and paired paracancerous tissue samples, among which five patients had tumor metastasis, and the other 5 patients had no tumor metastasis. Total RNA was isolated using a HiPure Total RNA Mini Kit (Magen Biotechnology, Guangzhou, China) following the manufacturer’s protocol. RNase R (Epicentre Biotechnologies, Madison, WI, USA) was used to digest the total RNA for 1 h after DNase I processing at 37 °C. After quantifying the sample RNA, the RNA library was prepared using a KAPA RNA Hyperprep Kit with Riborase (HMR) (Illumina, San Diego, CA, USA). Sequencing was performed using a Hiseq X10 system (Illumina) in PE150 mode. For analysis of circRNA-seq, sequencing readings were mapped to the Hg19 reference genome using STAR software (Dobin et al., 2013), and circRNA was identified using DCC tools (Cheng, Metge & Dieterich, 2016). The statistical power of this experimental design, calculated in RNASeq Power is 0.89. circRNAs that were differentially expressed in CRC were identified using the edgeR tools for screening of differentially expressed circRNA (Robinson, McCarthy & Smyth, 2010). Criteria for screening were a P-value of 0.05 and a fold change of ≥—2—. Differentially expressed circRNAs were visualized using differential distribution volcano plots and differential circRNA cluster analysis.

Quantitative real-time polymerase chain reaction (qRT-PCR)

Total RNA was extracted from tissues and cells using TRIzol. RNA was reverse transcribed into cDNA using an EasyScript First-Strand cDNA Synthesis SuperMix Kit. PCR was performed with a SYBR Green qPCR SuperMix Kit using the primers shown in Table 1 under the following conditions: 95 °C for 5 min; 40 cycles of 95 °C for 15 s and 60 °C for 32 s for plate reading; melting curve analysis temperature was 60 °C-−95 °C. The relative expression of genes was calculated according to the 2−△△CT formula.

Table 1 qPCR primer sequences.

Gene	Primer sequence (5′–3′)	
circMSH3-F	TAAATTAAGTGTGCAGGATGAC	
circMSH3-R	AGCCAAAGAGCAAATCACAG	
MSH3-F	ACGAGCACTCATGATGGAA	
MSH3-R	CCAAGAATCCCATGTGGTAA	
β-actin-F	GCATGGGTCAGAAGGATTCCT	
β-actin-R	TCGTCCCAGTTGGTGACGAT	
Notes.

Primer sequences for qPCR detection of circMSH3, MSH3 and internal reference genes.

RNA-protein pull-down

An Pierce™ Magnetic RNA-Protein Pull-Down Kit (20164; Thermo Fisher Scientific, Waltham, MA, USA) was used to detect interactions between circMSH3 and proteins. The circMSH3 probe solution (1 µg) was denatured at 90 °C for 2 min and incubated with pre-cooled RNA structure buffer to form RNA secondary structures. Next, streptavidin magnetic beads were incubated with the mixture at 25 °C for 30 min. The cells were lysed at 4 °C and centrifuged at 15,000× g for 15 min, and the supernatant was collected and mixed with the probe-bead mixture. An RNase inhibitor (5 µL) and poly(dI⋅ dC) (5 µL) were added, followed by incubation at 25 °C for 2 h and elution at 37 °C for 2 h. The supernatant was transferred into a new tube and used for protein mass spectrometry and western blotting. circMSH3 probe sequence were designed and generated by Ribo Bio (Guangzhou, China).

circMSH3: 5′-Biotin-UCCUUUCGACUCGAAUUCUG-3′;

Negative Control (NC): 5′-Biotin-UUCUCCGAACGUGUCACGUTT-3′.

Dual-luciferase reporter assay

Wild-type or mutant circMSH3 fragments were constructed and inserted downstream of the luciferase reporter gene in the pmirGLO plasmid (Promega, Madison, WI, USA). We transfected reporter plasmids into 293T cells using Lipofectamine 3000, followed by hsa-miR-409-3p, hsa-miR-942-5p, hsa-miR-1276, hsa-miR-513a-5p, and hsa-miR. The -7151-5p mimic was co-transfected with the reporter gene into 293T cells and the luciferase activity was measured 48 h later using a dual-luciferase reporter kit (Promega, Madison, WI, USA). The relative luciferase activity was normalized to the Renilla luciferase internal control.

Statistical analysis

IBM SPSS statistics 21.0 software (Chicago, IL, USA) was used for statistical analysis, GraphPad Prism 8.0 software (GraphPad Software, La Jolla, CA, USA) was used for graphing, and values were expressed as the mean ±standard deviation. The expression levels of circMSH3 in CRC and adjacent tissues were compared by a paired t-test. The relationship between the expression levels of circMSH3 in CRC tissue and the clinicopathological characteristics of the patients was compared between two groups using an independent samples t-test, and the comparison between multiple groups was performed using one-way analysis of variance. P-values <0.05 were considered statistically significant. The diagnostic performance of circMSH3 was analyzed using receiver operating characteristic (ROC) curves.

Results

Screening for differentially expressed circRNAs

In order to explore the expression characteristics of circRNAs in colorectal cancer tissues, RNA-seq results showed that most circRNAs originated from the exon region, chromosome 1-3 formed a large number of circRNAs, and the length of circRNA sequences was mainly distributed in 200 bp–500 bp (Figs. 1A–1C). circRNAs that were differentially expressed between teen pairs of metastatic and non-metastatic CRC groups were screened by high-throughput sequencing technology (Supplementary Materials 1). The results showed that compared with the non-metastatic CRC group, there were 87 upregulated circRNAs and 100 downregulated circRNAs in the metastatic CRC group (Fig. 1D). In the adjacent group, 176 circRNAs were up-regulated and 174 circRNAs were downregulated in the metastatic CRC group (Fig. 1E). Compared with the non-metastatic CRC adjacent group, more circRNAs were differentially expressed in the non-metastatic CRC group: the expression of 420 circRNAs was upregulated, and the expression of 616 circRNAs was downregulated (Fig. 1F). Taking the intersection of each group, there were 12 circRNAs that were differentially expressed in the tumor group, the paracancer group (N), and the non-metastatic and metastatic CRC groups (hsa_circ_0057123, hsa_circ_0000707, hsa_circ_0005704, hsa_circ_0004456, hsa_circ_0005379, hsa_circ_0130312, hsa_circ_0006672, hsa_circ_0001917, hsa_circ_0087641, hsa_circ_0003855, hsa_circ_0018992, and hsa_circ_0008230). At the same time, two circRNAs were differentially expressed between the metastatic CRC, non-metastatic CRC, and CRC adjacent groups (hsa_circ_0003761 [circMSH3] and hsa_circ_0133953) (Fig. 2A). hsa_circ_0133953 was discarded because its sequence was too long, and finally circMSH3 was selected for further research.

Figure 1 Screening of CRC-related circRNAs by high-throughput sequencing.

(A) Proportion of circRNA types. (B) Number of circRNAs distributed on each chromosome. (C) Length distribution of circRNAs. (D) Cluster heat map of differentially expressed circRNAs in non-metastatic CRC tissues and metastatic CRC tissue. Red represent the up −regulated circRNAs, blue represent the down −regulated circRNAs, and white represent the unchanged circRNAs. Each column represents one sample, and each row indicates a circRNA. E-F: The heat map of differentially expressed circRNAs in (metastatic vs paired normal) and (tumor vs paired normal).

Figure 2 Identification and characteristic analysis of circMSH3.

(A) Venn diagram of the intersection of the three differentially expressed circRNAs is shown. (B) Electrophoresis of the circMSH3 qRT-PCR product. (C) circMSH3 genome location, exon composition diagram, and Sanger sequencing. The results show the reverse splice site of circMSH3.

Verification of circMSH3 cyclability

circMSH3 was derived from exons 9, 10, 11, and 12 of the MSH3 gene with a length of 423 bp. To verify the circularity of circMSH3, we performed reverse Sanger sequencing of the PCR product. The sequencing results were consistent with the sequence of the circMSH3 circularization site in the CircBase database (Figs. 2B, 2C).

circMSH3 is a potential biomarker for the diagnosis of CRC

In 110 pairs of CRC and adjacent tissues, the relative expression of circMSH3 was significantly higher in CRC tissues (P < 0.001) (Fig. 3A). circMSH3 expression was significantly upregulated in the HCoEpiC epithelial cell line and colon cancer cell lines (HCT15,SW116,RKO,COLO205,HCT116,Caco2,LOVO,HT-29,SW480,SW620 and DLD-1), with the most significant upregulation in the SW480 cell line (Fig. 3B). circMSH3 and MSH3 gene expression was positively correlated in 30 CRC tissues, with a correlation coefficient of 0.64 (Fig. 3E). MSH3 gene and circMSH3 expression was upregulated in 30 CRC tissues (Figs. 3C–3D). Receiver Operating Characteristic curve (ROC) analysis showed that when circMSH3 was used for the diagnosis of CRC, its area under the curve (AUC) was 0.75, with a sensitivity and specificity of 70.9% and 66.4%, respectively (Fig. 3F). These results show that circMSH3 is a potential biomarker for the diagnosis of CRC.

Figure 3 Analysis of the diagnostic efficacy of circMSH3 in CRC tissues.

(A) qRT-PCR detection of the differential expression of circMSH3 in 110 pairs of CRC and adjacent normal tissues. (B) Detection of circMSH3 expression in a normal intestinal epithelial cell line (HCoEpiC) and cancer cell lines. (C) circMSH3 gene expression in 30 pairs of CRC and adjacent tissues. (D) MSH3 gene expression in 30 pairs of CRC and adjacent tissues. (E) Correlation analysis of circMSH3 and MSH3 mRNA expression in 30 pairs of CRC and adjacent tissues. (F) ROC curve analysis of the diagnostic efficacy of circMSH3 in CRC.

circMSH3 expression affects the distant metastasis of CRC

Using the clinical data of 110 patients, we analyzed the relationships between the expression levels of circMSH3 and the patients’ sex, age, tumor size, tumor infiltration degree (T), lymph node metastasis (N), distant metastasis (M), TNM stage, and differentiation. The results showed that circMSH3 expression was significantly correlated with distant metastasis in CRC patients (P = 0.043) but not with sex (P = 0.105), age (P = 0.526), tumor size (P = 0.953), degree of invasion (P = 0.509), lymph node metastasis (P = 0.596), TMN stage (P = 0.416), or degree of differentiation (P = 0.492) (Table 2). The results showed that circMSH3 expression affects the distant metastasis of CRC.

Table 2 Clinical characteristics of CRC patients.

Relationship between circMSH3 expression and the clinical characteristics of CRC patients.

	Patients (n)	Mean ± standard deviation	P- value	
Sex			0.105	
Male	65	6.5065 ± 9.5731		
Female	45	4.0151 ± 4.3182		
Age (years)			0.526	
≤60	50	4.9590 ± 6.7348		
>60	60	5.9275 ± 8.8324		
Tumor size (cm)			0.953	
≤5	82	5.4609 ± 8.2262		
>5	28	5.5646 ± 7.1213		
Degree of infiltration			0.509	
T1-T2	13	4.1154 ± 4.7397		
T3-T4	97	5.6711 ± 8.2622		
Lymphatic metastasis			0.596	
N0	55	5.0836 ± 5.8857		
N1–N2	55	5.8909 ± 9.5861		
Distant metastasis			0.043	
Yes	24	3.2988 ± 3.2988		
No	86	6.0049 ± 8.7412		
TNM			0.416	
I–II	46	5.1152 ± 6.2662		
III–IV	64	5.7547 ± 8.9733		
Differentiation			0.492	
Poor	10	7.4140 ± 7.4386		
Moderate	93	5.0347 ± 7.7308		
High	7	8.7471 ± 10.9829		

circMSH3-binding proteins and microRNAs (miRNAs)

Many studies have shown that circRNAs can act by binding to proteins or miRNAs (Hansen et al., 2013; Zang, Lu & Xu, 2020; Okholm et al., 2020). To explore the proteins and miRNA molecules that circMSH3 can bind to, we performed an RNA–protein pull-down assay combined with mass spectrometry and bioinformatics prediction combined with a dual-luciferase reporter assay. Biotin-labeled circMSH3 probe-captured proteins were identified by high performance liquid spectrometry (Figs. 4A, 4B). The Metascape website (Zhou et al., 2019) was used to perform function enrichment analysis of proteins, which showed that circMSH3 could bind to a variety of proteins involved in RNA transport and biological processes related to the cell cycle and cell junctions (Fig. 4D). These results suggest that RNA-splicing proteins may be involved in the formation of circMSH3, and circMSH3 may also regulate cell fate by binding directly to proteins involved in the cell cycle and cell junctions. The miRNAs that circMSH3 might bind to were predicted according to the starBase database, and the predictions were verified by a dual-luciferase reporter gene experiment. The results showed that circMSH3 could bind directly to hsa-miR-409-3p, hsa-miR-942-5p, and hsa-miR-1276, whereas no binding signal was detected for hsa-miR-513a-5p and hsa-miR-7151-5p (Fig. 5).

Figure 4 RNA pull-down assay analysis of circMSH3-binding proteins.

(A) The workflow of circMSH3 pull-down assay. (B) Silver staining electrophoresis before circMSH3-binding protein spectrum analysis. (C) Pull-down detection result for a HuR-positive probe. HuR pull-down was carried out by using the AR probe in the kit; the HuR antibody is supplied with the kit (36 kDa) and is used as a positive control in the same batch to exclude false negatives. D: Metascape tool for mass spectrometry identification of circMSH3-binding protein functional enrichment.

Figure 5 Dual-luciferase reporter gene detection of circMSH3 binding to miRNAs.

(A) Schematic diagram of circMSH3 binding miRNA sites predicted by bioinformatics; (B–F) Binding of 5 miRNAs to circMSH3 by dual luciferase assay. WT is the wild-type circMSH3 vector, while MUT is the mutant circMSH3 vector. Hela cells were transfected with the circMSH3-WT or -MUT vector. The cells were also transfected with each miRNA mimic and inhibitor, as well as the miRNA mimic control (NC) and the NC inhibitor control. A bioluminescence detector was used for dual fluorescence detection. Activity fold = (R / F) sample / (R / F) control, where F = firefly luciferase activity, and R = Renilla luciferase activity. *, p < 0.05. **, p < 0.01. ***, p < 0.001.

Discussion

circRNAs have been one of the most popular research topics in the RNA field in recent years. At the molecular level, circRNAs exert their biological functions mainly by acting as binding proteins for the “sponge adsorption” of miRNAs or proteins (Hansen et al., 2013; Zang, Lu & Xu, 2020). For example, hsa_circRNA_100084 can act as a sponge for hsa-miR-23a-5p to promote the expression of IGF2 in liver cancer cells, thereby promoting their proliferation, migration, and invasion (Yang et al., 2020). The circRNA-0008717/miR-203/Slug signaling axis promotes the migration, invasion, and metastasis of esophageal cancer cells, thus becoming a potential diagnostic and therapeutic target for esophageal cancer (Wang et al., 2020). A study of pancreatic cancer showed that circRNA-FOXK2 is significantly overexpressed in pancreatic cancer cells and tissues and can act as a sponge for miR-942, thereby promoting the expression of ANK1, GDNF, and PAX6, which in turn promote cell growth and migration via cell cycle progression and apoptosis. The downregulation of circRNA-FOXK2 can significantly inhibit the migration, invasion, liver metastasis, and tumor growth of pancreatic cancer cells (Wong et al., 2020). Recently, it was reported that circYap inhibits gene translation at the initial stage by affecting the mutual binding of PABP to the poly(A) tail and eIF4G to the 5′-cap of Yap mRNA (Wu et al., 2019). To date, the translational activity of circRNAs has been demonstrated in a variety of organisms (Lu et al., 2021).

We screened for circRNAs that were differentially expressed between five pairs of metastatic and non-metastatic CRC groups by high-throughput sequencing technology in the early stage. We selected the circRNA circMSH3 for further verification using qRT-PCR and expanded the clinical sample to 110 CRC samples, which showed that circMSH3 was significantly highly expressed in CRC tissues and cells. ROC curve analysis demonstrated that when circMSH3 was used for the diagnosis of CRC, its AUC was 0.75, with a sensitivity and specificity of 70.9% and 66.4%, respectively. Correlation analysis between circMSH3 and the clinical characteristics of CRC tissue samples showed that it was significantly associated with distant metastasis, indicating that circMSH3 can be used as a potential biomarker for the diagnosis of CRC and prediction of distant metastasis.

In terms of mechanism, we initially explored the proteins and miRNAs that circMSH3 can bind to. According to an RNA–protein pull-down assay combined with mass spectrometry, we found that circMSH3 can bind to a variety of proteins involved in splicing, methylation, RNA transport, and biological processes related to the cell cycle and cell junctions (Fig. 4D). These results suggest that RNA-splicing proteins may be involved in the formation of circMSH3, and that circMSH3 may also interact directly with proteins involved in the cell cycle and cell junctions. Bioinformatics prediction combined with a dual-luciferase reporter gene assay to detect the possible miRNAs bound by circMSH3, showed that circMSH3 can bind directly to hsa-miR-409-3p, hsa-miR-942-5p, and hsa-miR-1276. These results indicate that circMSH3 may exert its biological functions by binding to various proteins and miRNAs. Of course, if the exact function and molecular mechanism of circMSH3 in the distant metastasis of CRC are to be determined, further experiments are needed.

It should be recognized that this study also has the following limitations. First of all, limited by the number of collected colorectal cancer clinical samples, we will collect more samples for the development of circMSH3 as a diagnostic marker in the future. Secondly, the results of luciferase reporter gene experiments and RNA pulldown experiments in the previous period showed that circMSH3 can bind multiple miRNAs and proteins. These clues suggest that circMSH3 may play an important role in colorectal cancer and is an important potential therapeutic target. It will be promising to further explore the molecular mechanism and cellular function of circMSH3 in vivo and in vitro, which will be of value in elucidating the specific mechanism of circMSH3 in colorectal cancer.

In conclusion, the findings of this article show that circMSH3 is a potential marker for the diagnosis of CRC, may have important molecular functions in the occurrence and development of CRC, and can be used as a potential therapeutic target.

Supplemental Information

Supplemental Information 1 Differentially expressed circRNAs

10 pairs of colorectal cancer patients compared between groups, differentially expressed circRNAs.

Click here for additional data file.

Supplemental Information 2 Original result image of Fig. 2B

Fig. 2B corresponds to the original photo of qPCR product electrophoresis gel electrophoresis of circMSH3

Click here for additional data file.

Supplemental Information 3 The original photo corresponding to Fig. 4C

The photo of the original film corresponding to the WB electrophoresis results of the positive control HuR in the pulldown experiment

Click here for additional data file.

Supplemental Information 4 Fig. 5 corresponds to the original experimental result file

The original result files including the circMSH3 binding miRNAs site prediction, dual fluorescent reporter gene carrier sequencing results and signal detection.

Click here for additional data file.

Additional Information and Declarations

Competing Interests

Author Contributions

Human Ethics

DNA Deposition

Data Availability

The authors declare there are no competing interests.

Jian Shen conceived and designed the experiments, performed the experiments, analyzed the data, prepared figures and/or tables, and approved the final draft.

Yu Min conceived and designed the experiments, performed the experiments, analyzed the data, prepared figures and/or tables, and approved the final draft.

Jingen Luo analyzed the data, authored or reviewed drafts of the article, and approved the final draft.

Xingkui Tang analyzed the data, authored or reviewed drafts of the article, and approved the final draft.

Zeping Han performed the experiments, analyzed the data, prepared figures and/or tables, and approved the final draft.

Wenfeng Luo performed the experiments, analyzed the data, prepared figures and/or tables, and approved the final draft.

Fangmei Xie performed the experiments, analyzed the data, prepared figures and/or tables, and approved the final draft.

Mingrong Cao conceived and designed the experiments, authored or reviewed drafts of the article, and approved the final draft.

Taicheng Zhou conceived and designed the experiments, prepared figures and/or tables, authored or reviewed drafts of the article, and approved the final draft.

Jinhua He conceived and designed the experiments, authored or reviewed drafts of the article, and approved the final draft.

The following information was supplied relating to ethical approvals (i.e., approving body and any reference numbers):

The Guangzhou Panyu District Central Hospital gave approval to carry out the study within its facilities.

The following information was supplied regarding the deposition of DNA sequences:

The raw data of RNA-seq high-throughput sequencing is available at SRA: PRJNA926691.

The following information was supplied regarding data availability:

The raw data of RNA-seq high-throughput sequencing is available at SRA: PRJNA926691.

The raw measurements are available in the Supplemental Files.

https://www.ncbi.nlm.nih.gov/bioproject/PRJNA926691

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
