# Peer review of "circMSH3 is a potential biomarker for the diagnosis of colorectal cancer and affects the distant metastasis of colorectal cancer"

_PeerJ, doi:10.7717/peerj.16297_

## Round 0.1 · original submission · Minor Revisions

We hope this email finds you well. We are writing to express our gratitude for your submission of the paper titled "circMSH3 is a potential biomarker for the diagnosis of colorectal cancer and affects the distant metastasis of colorectal cancer" to our esteemed journal. We appreciate the time and effort you invested in preparing this manuscript.
We are pleased to inform you that the review process has been completed, and we received feedback from the reviewers assigned to your paper. Both reviewers have recognized the potential of your work to contribute significantly to the field and have provided valuable insights to enhance the manuscript further. While they consider your paper worthy of publication, they have raised a few minor concerns that need to be addressed to ensure its suitability for publication in our journal.
We kindly request that you address each concern in a point-by-point manner and provide a clear and concise response for the benefit of the reviewers.

·

Basic reporting

The authors have done a good job in providing a good context on the background and importance of cirRNAs in the diagnosis of CRC with well outlined abstract and introduction section.

Some minor recommended edits regarding acronyms, referencing & figures:
1) Line 111: Materials and methods - RNA protein pull down; The paper uses the acronym NC first time & is not written in full.
2) Line 132: Teen pairs - likely a typo mistake
3) Line 140: Figure 1G-I referenced, but not provided in the supplemental data set.
4) Line 151: Figure 2A incorrectly referenced on line 151. It should be referenced at the end of line 148.
5) Line 162: Acronym ROC not mentioned in full.
6) Line 183 and line 221: The figure referenced is 4C - it should be 4D.

Experimental design

No comments

Validity of the findings

The paper highlights that although circRNAs have been shown to play a role in tumorigenesis and development, the key circRNAs that regulate CRC progression and their molecular mechanisms remain unclear. Identifying the key circRNAs in CRC and understanding their functions and molecular mechanisms can provide new molecular markers for clinical diagnosis and treatment of CRC and help design therapeutic strategies to inhibit metastasis. The explanations provided in the Methods and Results section were quite thorough, including a clear thought process on the why those methods were selected.

Comments:
1) Are the authors confident on concluding that cirMSH3 is a potential biomarker just on the basis of data collected from 110 patients? This clearly reflects as one of the limitations & it should be mentioned.

2) How were the CRC cell lines selected? Maybe add some references.

3) The authors also need to make suggestions for future research including the limitations.

Reviewer 2 ·

Basic reporting

1 For figure 1 DEF, please explain what is the meaning of the green and red color of the heatmap in figure caption.

2. For the errorbar figures, please indicate the significant level of “*” in figure captions.

3. Figure 5 B, please remove a letter “C” from the figure.

Experimental design

In section “circRNA sequencing (circRNA-seq)”, line 90, please clarify that the criteria “P-value of <0.05” in your study is the original P-value or adjusted P-value. If the original P-value was used, please justify why the adjusted P-value was not used.

Validity of the findings

The authors is diclosing that CircMSH3 holds promise as a diagnostic biomarker for colorectal cancer (CRC) and plays a role in CRC's distant metastasis. Several RNA-binding proteins have been found to interact with circMSH3, and circMSH3, in turn, binds to miR-1276, miR-942-5p, and miR-409-3p, consequently influencing the expression of circMSH3. The results and conclusions are relatively promising.

Additional comments

NA

---

## Round 0.2 · accepted · Accept

To facilitate this process, please consider some of the points by the reviewer related to the references, this will help enhance the clarity and flow of your manuscript. Additionally, we kindly request that you submit the final version of your manuscript with these changes.

·

Basic reporting

No comments

Experimental design

No comment

Validity of the findings

No comment

Additional comments

Dear Authors,

I would like to express my heartfelt appreciation for the outstanding work you have done on this manuscript. Your dedication and hard work are evident, and I fully endorse the publication of this manuscript.

Nevertheless, I would like to draw your attention to a few minor discrepancies that need your attention. I noticed that there seems to be an inconsistency in the references within the Peerj_response document_review and the editor's notes.

While the revisions you've made in response to the suggestions are reflective, it appears there might be a slight mix-up with the line references in your responses.

For example: "We have added the full name 'Negative Control (NC)' in Page 5, Line 137 of the revised manuscript." However, upon my review, this change appears to have been made on line 126, not 137 as mentioned.

I kindly request you to review these references once more. This will ensure that all the line references correspond accurately to the changes made. If you plan to publish the reviewers' comments and responses, it would be beneficial to ensure the references are correctly aligned.

Reviewer 2 ·

Basic reporting

The revised paper expressed their thoughts and ideas clearly and unambiguously with sufficient background and literature references provided. The revised manuscript has professional structure including good figures and tables.

Experimental design

The authors revised the manuscript related to experimental design based on reviewers' comments. Currently, the research questions are well defined and relevant and meaningful. The statistical analysis is valid and rigorous to make sure the investigation was performed rigorously to a high technical and ethical standard. Methods were described with sufficient details to replicate.

Validity of the findings

Since the manuscript was refined according to the reviewers' comments, the findings in this manuscript are valid. Sufficient underlying data have been provided which are robust, statistically sound. The conclusions are well stated and linked to original research question.

Additional comments

I recommend to accept.